# Particle-Based Score Estimation for State Space Model Learning in Autonomous Driving

**Angad Singh**[*]
Waymo Research
singhangad@waymo.com

**Omar Makhlouf**[*]
Waymo Research
makhlouf@waymo.com

**Maximilian Igl**
Waymo Research
migl@waymo.com

**Joao Messias**
Waymo Research
messiasj@waymo.com

**Arnaud Doucet**
DeepMind
arnauddoucet@deepmind.com

**Shimon Whiteson**
Waymo Research
shimonw@waymo.com

**Abstract:** Multi-object state estimation is a fundamental problem for robotic applications where a robot must interact with other moving objects. Typically, other objects' relevant state features are not directly observable, and must instead be inferred from observations. Particle filtering can perform such inference given approximate transition and observation models. However, these models are often unknown a priori, yielding a difficult parameter estimation problem since observations jointly carry transition and observation noise. In this work, we consider learning maximum-likelihood parameters using particle methods. Recent methods addressing this problem typically differentiate through time in a particle filter, which requires workarounds to the non-differentiable resampling step, that yield biased or high variance gradient estimates. By contrast, we exploit Fisher's identity to obtain a particle-based approximation of the score function (the gradient of the log likelihood) that yields a low variance estimate while only requiring stepwise differentiation through the transition and observation models. We apply our method to real data collected from autonomous vehicles (AVs) and show that it learns better models than existing techniques and is more stable in training, yielding an effective smoother for tracking the trajectories of vehicles around an AV.

**Keywords:** Autonomous Driving, Particle Filtering, Self-supervised Learning

## 1 Introduction

Multi-object state estimation is a fundamental problem in settings where a robot must interact with other moving objects, since their state is directly relevant for decision making. Typically, other objects' relevant state features are not directly observable. Instead, the robot must infer them from a stream of observations it receives via a perception system. For example, an autonomous vehicle (AV) selects actions based on the state of nearby road users. However, such road users are only partially observed, owing to limited field of view, occlusions, and imperfections in the AV's sensors and perception systems. Such partial observability negatively affects many downstream tasks in a robot's behavioural stack that depend on observations, e.g., action planning.

Addressing partial observability requires sequential state estimation, to which Bayesian filtering offers a generic probabilistic approach. In particular, sequential Monte Carlo methods, also known as particle filtering, have been successfully applied to state estimation in many robotics applications [1]. However, Bayesian filters require models that reasonably approximate the transition and observation models of a state-space model (SSM). In some special cases, these models can be derived analytically from first principles, e.g., when the physical dynamics are well understood, or by modeling a sensor's physical characteristics. In many real-world applications, however, these models cannot be specified analytically. For example, the transition model may encode complicated motion dynamics and environmental physics. In multi-agent settings, other agents' behaviour must also be modelled.

---

[*]These authors contributed equally to this work.

6th Conference on Robot Learning (CoRL 2022), Auckland, New Zealand.

Modelling observations is also difficult. Modern perception systems often involve multiple stages and combine information from multiple sensors, making observation models practically impossible to specify by hand. By contrast, collecting observations from a robotic system is relatively easy and cheap. We are interested, therefore, in algorithms that can leverage such observations to learn transition and observation models in a self-supervised fashion, and yield an effective particle smoother. Learned transition and observation models can also be independently useful for other applications, such as the evaluation of AVs by simulating realistic observations.

In this work, we propose Particle Filtering-Based Score Estimation using Fisher's Identity (PF-SEFI), a method for jointly learning maximum-likelihood parameters of both the transition and observation models, that is applicable to a wide class of SSMs. Unlike many recently proposed methods [2, 3, 4, 5, 6, 7], our approach avoids differentiable approximations of the resampling step. We achieve this by revisiting a methodology originally proposed in statistics [8, 9] that relies on a particle approximation of the score, i.e., the gradient of the log likelihood of observation sequences, obtained through Fisher's identity. This only requires differentiating through the transition and observation models. Unfortunately, a direct particle approximation of this identity provides a high variance estimate of the score. While [8] propose an alternative low variance estimate, it admits a $\mathcal{O}(N^2)$ cost, where $N$ is the number of particles. Furthermore, these methods compute and store the gradient of the marginal log-likelihood with respect to model parameters for each particle. This requires computing Jacobian matrices, which are slow to compute using automatic differentiation tools such as TensorFlow and PyTorch [10, 11] which rely on Jacobian-vector products. This makes these methods impractical for large models. By contrast, PF-SEFI is a simple scalable $\mathcal{O}(N)$ variant with only negligible bias. PF-SEFI marginalises over particles before computing gradients, allowing automatic differentiation tools to make use of efficient Jacobian-vector product operations, making it significantly faster and allowing us to scale to larger models. To the best of our knowledge, previous particle methods estimating the score have been limited to SSMs with few parameters, whereas we apply PF-SEFI to neural network models with thousands of parameters.

We apply PF-SEFI to jointly learn transition and observation models for tracking multiple objects around an AV, using a large set of noisy trajectories, containing almost 10 hours of road-user trajectories observed by an AV. We show that PF-SEFI learns an SSM that yields an effective object tracker as measured by average displacement and yaw errors. We compare the learned observation model to one trained through supervised learning on a dataset of manually labelled trajectories, and show that PF-SEFI yields a better model (as measured by log-likelihood on ground-truth labels) even though it requires no labels for training. Finally, we compare PF-SEFI to a number of existing particle methods for jointly learning transition and observation models and show that it learns better models and is more stable to train.

## 2   Related Work

Particle filters are widely used for state estimation in non-linear non-Gaussian SSMs where no closed form solution is available; see e.g., [12] for a survey. The original bootstrap particle filter [13] samples at each time step using the transition density particles that are then reweighted according to their conditional likelihood, which measures their "fitness" w.r.t. to the available observation. Particles with low weights are then eliminated while particles with high weights are replicated to focus computational efforts into regions of high probability mass. Compared to many newer methods, such as the auxiliary particle filter [14], the bootstrap particle filter only requires sampling from the transition density, not its evaluation at arbitrary values, which is not possible for the compositional transition density used in this work.

In most practical applications, the SSM has unknown parameters that must be estimated together with the latent state posterior (see [9] for a review). Simply extending the latent space to include the unknown parameters suffers from insufficient parameter space exploration [15]. While particle filters can estimate consistently the likelihood for fixed model parameters, a core challenge is that the such estimated likelihood function is discontinuous in the model parameters due to the resampling step, hence complicating its optimization; see e.g. [6, Figure 1] for an illustration.

Instead, the score vector can be computed using Fisher's identity [8]. However, as shown in [8], performance degrades quickly for longer sequences if a standard particle filter is used, due to the path degeneracy problem: repeated resampling of particles and their ancestors will leave few or even

just one remaining ancestor path for earlier timesteps, resulting in unbiased, but very high variance estimates. Methods for overcoming this limitation exist [8, 16, 17], but with requirements making them unsuitable in this work. Poyiadjis et al. [8] store gradients separately for each particle, making this approach infeasible for all but the smallest neural networks. Ścibior and Wood [17] propose an improved implementation with lower memory requirements by smartly using automatic differentiation. However, their approach still requires storing a computation graph whose size scales with $\mathcal{O}(N^2)$ as the transition density for each particle *pair* must be evaluated during the forward pass. Both previous methods' computational complexity also scales quadratically with the number of particles, $N$, which is problematic for costly gradient backpropagation through large neural networks. Lastly, Olsson and Westerborn [16] require evaluation of the transition density for arbitrary values, which our compositional transition model does not allow. Instead, in this work, we show that fixed-lag smoothing [18, 19] is a viable alternative to compute the score function of large neural network models in the context of extended object tracking.

There is extensive literature on combining particle filters with learning complex models such as neural networks [2, 3, 4, 5, 6, 20, 21, 22, 23, 24]. In contrast to our work, they make use of a learned, data-dependent proposal distribution. However, for parameter estimation, they rely on differentiation of an estimated lower bound (ELBO). Due to the non-differentiable resampling step, this gradient estimation has either extremely high variance or is biased if the high variance terms are simply dropped, as in [2, 3, 4]. As we show in Section 5, this degrades performance noticeably. A second line of work proposes soft resampling [5, 20, 21], which interpolates between regular and uniform sampling, thereby allowing to trade off variance reduction through resampling with the bias introduced by ignoring the non-differentiable component of resampling. Lastly, Corenflos et al. [6] make the resampling step differentiable by using entropy-regularized optimal transport, also inducing bias and a $\mathcal{O}(N^2)$ cost.

Extended object tracking [25] considers how to track objects which, in contrast to "small" objects [26], generate multiple sensor measurements per timestep. Unlike in our work, transition and measurement models are assumed to be known or to depend on only a few learnable parameters. Similar to our work, the measurement model proposed in [27] assumes measurement sources lying on a rectangular shape. However, our model is more flexible, for example, allowing non-zero probability on all four sides simultaneously.

## 3  State-Space Models and Particle Filtering

### 3.1  State-Space Models

A SSM is a partially observed discrete-time Markov process with initial density, $x_0 \sim \mu(\cdot)$, transition density $x_t|x_{t-1} \sim f_\theta(\cdot|x_{t-1})$, and observation density $y_t|x_t \sim g_\theta(\cdot|x_t)$, where $x_t$ is the latent state at time $t$ and $y_t$ the corresponding observation. The joint density of $x_{0:T}, y_{0:T}$ satisfies:

$$p_\theta(x_{0:T}, y_{0:T}) = \mu(x_0)g_\theta(y_0|x_0) \prod_{t=1}^{T} f_\theta(x_t|x_{t-1})g_\theta(y_t|x_t). \tag{1}$$

Given this model, we are typically interested in inferring the states from the data by computing the filtering and one-step ahead prediction distributions, $\{p(x_t|y_{0:t})\}_{t\in 0,\dots,T}$ and $\{p(x_{t+1}|y_{0:t})\}_{t\in 0,\dots,T-1}$ respectively, and more generally the joint distributions $\{p(x_{0:t}|y_{0:t})\}_{t\in 0,\dots,T}$ satisfying

$$p_\theta(x_{0:t}|y_{0:t}) = \frac{p_\theta(x_{0:t}, y_{0:t})}{p_\theta(y_{0:t})}, \qquad p_\theta(y_{0:T}) = \int p_\theta(x_{0:T}, y_{0:T})\mathrm{d}x_{0:T}. \tag{2}$$

Additionally, to estimate parameters, we would also like to compute the marginal log likelihood:

$$\ell_T(\theta) = \log p_\theta(y_{0:T}) = \log p_\theta(y_0) + \sum_{t=1}^{T} \log p_\theta(y_t|y_{0:t-1}), \tag{3}$$

where $p_\theta(y_0) = \int g_\theta(y_0|x_0)\mu(x_0)\mathrm{d}x_0$ and $p_\theta(y_t|y_{0:t-1}) = \int g_\theta(y_t|x_t)p_\theta(x_t|y_{0:t-1})\mathrm{d}x_t$ for $t \geq 1$.

For non-linear non-Gaussian SSMs, these posterior distributions and the corresponding marginal likelihood cannot be computed in closed form.

## 3.2 Particle Filtering

Particle methods provide non-parametric and consistent approximations of these quantities. They rely on the combination of importance sampling and resampling steps of a set of $N$ weighted particles $(x_t^i, w_t^i)$, where $x_t^i$ denotes the values of the $i^{\text{th}}$ particle at time $t$ and $w_t^i$ is corresponding weight satisfying $\sum_{i=1}^{N} w_t^i = 1$. We focus on the bootstrap particle filter, shown in Algorithm 1, which samples particles according to the transition density. Let $k \sim \text{Cat}(\alpha_1, ..., \alpha_N)$ denote the categori-

---

**Algorithm 1** Bootstrap Particle Filter

---

Sample $X_0^i \overset{\text{i.i.d.}}{\sim} \mu(\cdot)$ for $i \in [N]$ and set $\hat{\ell}_0(\theta) \leftarrow \log\left(\frac{1}{N}\sum_{i=1}^{N} g_\theta(y_0|x_0^i)\right)$.

For $t = 1, ..., T$

    1. Compute weights $w_{t-1}^i \propto g_\theta(y_{t-1}|x_{t-1}^i)$ with $\sum_{i=1}^{N} w_{t-1}^i = 1$.

    2. Sample $a_{t-1}^i \sim \text{Cat}(w_{t-1}^1, ..., w_{t-1}^N)$ then $x_t^i \sim f_\theta(\cdot|x_{t-1}^{a_{t-1}^i})$ for $i \in [N]$.

    3. Set $x_{0:t}^i \leftarrow (x_{0:t-1}^{a_{t-1}^i}, x_t^i)$ for $i \in [N]$ and $\hat{\ell}_t(\theta) \leftarrow \hat{\ell}_{t-1}(\theta) + \log\left(\frac{1}{N}\sum_{i=1}^{N} g_\theta(y_t|x_t^i)\right)$.

---

cal distribution with $N$ categories, where the probability of the $k$ taking the $i^{\text{th}}$ category is $\alpha_i$. At any time $t$, this algorithm produces particle approximations

$$\hat{p}_\theta(x_{0:t}|y_{0:t}) = \sum_{i=1}^{N} w_t^i \delta_{x_{0:t}^i}(x_{0:t}), \qquad \hat{\ell}_t(\theta) = \sum_{t=0}^{T} \log\left(\frac{1}{N}\sum_{i=1}^{N} g_\theta\left(y_t|x_t^i\right)\right), \qquad (4)$$

of $p_\theta(x_{0:t}|y_{0:t})$ and $\ell_t(\theta) = \log p_\theta(y_{0:t})$, where $\delta_\alpha$ is the Dirac delta distribution centred at $\alpha$. Step 2 resamples, discarding particles with small weights while replicating those with large weights before evolving according to the transition density. This focuses computational effort on the "promising" regions of the state space. Unfortunately, resampling involves sampling $N$ discrete random variables at each time step and as such produces estimates of the log likelihood that are not differentiable w.r.t. $\theta$ as illustrated in [6, Figure 1].

While the resulting estimates are consistent as $N \to \infty$ for any fixed time $t$ [28], this does not guarantee good practical performance. Fortunately, under regularity conditions the approximation error for the estimate $\hat{p}_\theta(x_t|y_{0:t})$ and more generally $\hat{p}_\theta(x_{t-L+1:t}|y_{0:t})$ for a fixed lag $L \geq 1$ as well as $\log p_\theta(y_{0:t})/t$ does not increase with $t$ for fixed $N$. However, this is not the case for the joint smoothing approximation because successive resampling means that $\hat{p}_\theta(x_{0:L}|y_{0:t})$ is eventually approximated by a single unique particle for large enough $t$, a phenomenon known as path degeneracy; see e.g. [12, Section 4.3].

## 4 Score Estimation using Particle Methods

To estimate the parameters $\theta$ of a given SSM (1) along with a dataset of observations $y_{0:T}$, we want to maximise via gradient ascent the marginal log likelihood in (3). However, the gradient of the marginal log likelihood, i.e., the *score function*, is intractable. As explained in Section 2, automatic differentiation through the filter is difficult due to the non-differentiable resampling step.

### 4.1 Score Function Using Fisher's Identity

We leverage here instead Fisher's identity [8] for the score to completely side-step the non-differentiability problem. This identity shows that

$$\nabla_\theta \ell_T(\theta) = \int \nabla_\theta \log p_\theta(x_{0:T}, y_{0:T})\, p_\theta(x_{0:T}|y_{0:T}) \mathrm{d}x_{0:T}, \qquad (5)$$

i.e., the score is the expectation of $\nabla_\theta \log p_\theta(x_{0:T}, y_{0:T})$ under the joint smoothing distribution $p_\theta(x_{0:T}|y_{0:T})$. Plugging in (1), the score function can be simplified to

$$\begin{aligned}
\nabla_\theta \ell_T(\theta) &= \sum_{t=0}^{T} \int \nabla_\theta \log g_\theta(y_t|x_t)\, p_\theta(x_t|y_{0:T}) \mathrm{d}x_t \\
&\quad + \sum_{t=1}^{T} \int \nabla_\theta \log f_\theta(x_t|x_{t-1})\, p_\theta(x_{t-1:t}|y_{0:T}) \mathrm{d}x_{t-1:t}. \qquad (6)
\end{aligned}$$

## 4.2 Particle Score Approximation

The identity (6) shows that we can simply estimate the score by plugging particle approximations of the marginal smoothing distributions $p(x_{t-1:t}|y_{0:T})$ into (6). This identity makes differentiating through time superfluous and thereby renders the use of differentiable approximations of resampling unnecessary. However, as discussed in Section 3.2, naive particle approximations of the smoothing distribution's marginals, $p_\theta(x_t|y_{0:T})$ and $p_\theta(x_{t-1:t}|y_{0:T})$, suffer from path degeneracy. To bypass this problem, [8, 17] propose an $\mathcal{O}(N^2)$ method inspired by dynamic programming. We propose here a simpler and computationally cheaper method that relies on the following fixed-lag approximation of the fixed-interval smoothing distribution, which states that for $L \geq 1$ large enough,

$$p_\theta(x_{t-1:t}|y_{0:T}) \approx p_\theta\left(x_{t-1:t}|y_{0:\min\{t+L,T\}}\right). \tag{7}$$

This approximation simply assumes that observations after time $t + L$ do not bring further information about the states $x_{t-1}, x_t$. This is satisfied for most models and the resulting approximation error decreases geometrically fast with $L$ [19]. The benefit of this approximation is that the particle approximation of $p_\theta\left(x_{t-1:t}|y_{0:\min\{t+L,T\}}\right)$ does not suffer from path degeneracy and is a simple byproduct of the bootstrap particle filtering of Algorithm 1; e.g., for $t + L < T$ we consider the particle approximation $\hat{p}_\theta(x_{0:t+L}|y_{0:t+L}) = \sum_{i=1}^N w_{t+L}^i \delta_{x_{0:t+L}^i}(x_{0:t+L}^i)$ obtained at time $t + L$ and use its corresponding marginals in $x_{t-1}, x_t$ and $x_t$ to integrate respectively $\nabla_\theta \log f_\theta(x_t|x_{t-1})$ and $\nabla_\theta \log g_\theta(y_t|x_t)$. For $t + L \geq T$, we just consider the marginals in $x_{t-1}, x_t$ and $x_t$ of $\hat{p}_\theta(x_{0:T}|y_{0:T})$. So finally, we consider the estimate,

$$\widehat{\nabla_\theta \ell_T}(\theta) = \sum_{t=0}^T \int \nabla_\theta \log g_\theta(y_t|x_t) \, \hat{p}_\theta(x_t|y_{0:\min\{t+L,T\}}) \mathrm{d}x_t$$

$$+ \sum_{t=1}^T \int \nabla_\theta \log f_\theta(x_t|x_{t-1}) \, \hat{p}_\theta(x_{t-1:t}|y_{0:\min\{t+L,T\}}) \mathrm{d}x_{t-1:t}. \tag{8}$$

## 4.3 Score Estimation with Deterministic, Differentiable, Injective Motion Models

We have described a generic method to approximate the score using particle filtering techniques. For many applications, however, the transition density function, $f_\theta(x_t|x_{t-1})$, is the composition of a *policy*, $\pi_\theta(a_t|x_{t-1})$, which characterises the action distribution conditioned on the state, and a potentially complex but deterministic, differentiable, and injective *motion model*, $\tau : \mathbb{R}^{n_x} \times \mathbb{R}^{n_a} \to \mathbb{R}^{n_x}$ where $n_a < n_x$, which characterises kinematic constraints such that $x_t = \tau(x_{t-1}, a_t) = \bar{\tau}_{x_{t-1}}(a_t)$. Under such a composition, the transition density function on the induced manifold $\mathcal{M}_{x_{t-1}} = \{\bar{\tau}_{x_{t-1}}(a_t) : a_t \in \mathbb{R}^{n_a}\}$ is thus obtained by marginalising out the latent action variable, i.e.,

$$f_\theta(x_t|x_{t-1}) = \mathbb{I}(x_t \in \mathcal{M}_{x_{t-1}}) \int \delta(x_t - \bar{\tau}_{x_{t-1}}(a_t)) \, \pi_\theta(a_t|x_{t-1}) \mathrm{d}a_t. \tag{9}$$

It is easy to sample from this density but it is intractable analytically if the motion model is only available through a complex simulator or if it is not invertible. This precludes the use of sophisticated proposal distributions within the particle filter. Additionally, even if it were known, one cannot use the $\mathcal{O}(N^2)$ smoothing type algorithms developed in [8, 16] as the density is concentrated on a low-dimensional manifold [29]. This setting is common in mobile robotics, in which controllers factor into policies that select actions and motion models that determine the next state. Indeed, this is precisely the case in our application (see Section 5). Learning the corresponding SSM reduces to learning the parameters $\theta$ of the policy, $\pi_\theta(a_t|x_{t-1})$, and the observation model, $g_\theta(y_t|x_t)$. Thankfully, even if the explicit form of the motion model is unknown, we can still compute $\nabla \log f_\theta(x_t|x_{t-1})$ as required by the score estimate (8).

**Lemma 4.1.** *For any $x \in \mathbb{R}^{n_x}$, let $\tau_x : \mathbb{R}^{n_a} \to \mathbb{R}^{n_x}$ where $n_a < n_x$ be a smooth and injective mapping. Then, for any fixed $x_{t-1}$ and $x_t \in \mathcal{M}_{x_{t-1}}$, the gradient of the transition log density, i.e., $\nabla_\theta \log f_\theta(x_t|x_{t-1})$, reduces to the gradient of the policy log density, i.e., $\nabla_\theta \log \pi_\theta(a_t|x_{t-1})$, where $a_t$ is the unique action that takes $x_{t-1}$ to $x_t$.*

*Proof.* For $x_{t-1}$ and $x_t \in \mathcal{M}_{x_{t-1}}$, we denote by $J[\bar{\tau}_{x_{t-1}}](\bar{\tau}_{x_{t-1}}^{-1}(x_t)) \in \mathbb{R}^{n_x \times n_a}$ the rectangular Jacobian matrix and write $a_t = \bar{\tau}_{x_{t-1}}^{-1}(x_t)$, i.e., this is the unique action such $\bar{\tau}_{x_{t-1}}(a_t) = x_t$. By a

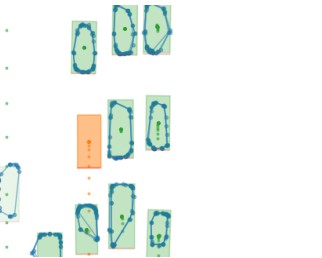 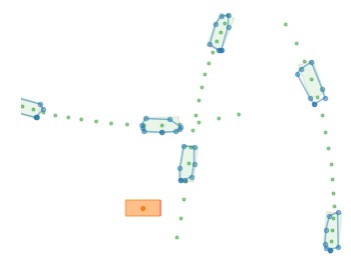

**(a)** Observations from the real data.      **(b)** Observations from the synthetic data.

**Figure 1:** Observations from real and synthetic data. The AV (orange) observes a set of 2D points (blue) forming a polygon around the road users. The true bounding boxes of each road user (green) were manually labelled in the real data, and pre-determined while constructing the synthetic data.

standard result from differential geometry [30, 31], the transition density (9) satisfies

$$f_\theta(x_t|x_{t-1}) = \pi_\theta(a_t|x_{t-1}) \left| \det J[\bar\tau_{x_{t-1}}]^{\mathrm{T}}(a_t) J[\bar\tau_{x_{t-1}}](a_t) \right|^{-1/2} \mathbb{I}(x_t \in \mathcal{M}_{x_{t-1}}). \qquad (10)$$

It follows directly that $\nabla_\theta \log f_\theta(x_t|x_{t-1}) = \nabla_\theta \log \pi_\theta(a_t|x_{t-1})$. ∎

Indeed for the marginals $\hat p_\theta\left(x_{t-1:t}|y_{0:\min\{t+L,T\}}\right)$, we can store the actions corresponding to transitions $x_{t-1} \to x_t$ during filtering, and it follows that for the class of SSMs described above, the score estimate reduces to:

$$\widehat{\nabla_\theta \ell_T}(\theta) = \sum_{t=0}^{T} \int \nabla_\theta \log g_\theta(y_t|x_t)\, \hat p_\theta(x_t|y_{0:\min\{t+L,T\}}) \mathrm{d}x_t$$

$$+ \sum_{t=1}^{T} \int \nabla_\theta \log \pi_\theta(a_t|x_{t-1})\, \hat p_\theta(x_{t-1:t}|y_{0:\min\{t+L,T\}}) \mathrm{d}x_{t-1:t}, \qquad (11)$$

where we use Lemma 4.1 to replace the gradient of the transition log density with the gradient of the policy log density in (8), and where $a_t$ is the action sampled to go from $x_{t-1}$ to $x_t$.

## 5 Experiments

**Problem Setting.** Our experiments focus on the problem of state estimation of observed road users (in particular other vehicles) from the viewpoint of an AV, which involves the estimation of 2D poses from an observed sequence of 2D convex polygons in a "bird's eye view" (BEV) constructed from LiDAR point clouds at each time step. For these experiments, we assume that the size of the observed objects, the pose of the AV, and the association of observations with their corresponding objects are known a priori. Some observations (and their corresponding states) are shown in Figure 1a. Here, the observation model must learn to describe the likelihood of 2D points around the periphery of the observed road user (see [25] for a review on such models), while the transition model must learn to describe driving behaviour. We use a feed-forward neural network to parameterise our observation model, where we provide it with range, bearing, and relative bearing from the viewpoint of the corresponding AV as features (Appendix A), and factor our transition model into a deterministic and differentiable motion model based on Ackermann dynamics [32] (Appendix B.1), and a policy parameterised by another feed-forward neural network (Appendix B.2).

**Baselines, Datasets, and Metrics.** We compare the quality of the models learned using PF-SEFI (our method), DPF-SGR [17], PFNET [5], and differentiating through a vanilla PF (ignoring the bias introduced by resampling). We evaluate our method (and the baselines) on real data collected from an AV in an urban environment, equipped with LiDARs, cameras, and radar sensors. All sensors were used to associate LiDAR points to their corresponding objects, and the observations shown in Figure 1a were obtained via a convex hull computation of the associated LiDAR points. In addition to using real data, we generate two synthetic datasets (with 25 and 50 step trajectories), using a hand-crafted policy, and an observation model trained using supervised learning on manually labelled trajectories (see Appendix A and B for more details). Example observations are shown in Figure 1b. Unlike with real data, where the true models are unknown, synthetic datasets allow us

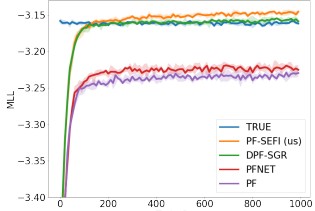 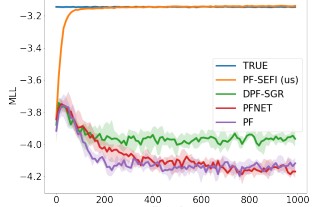 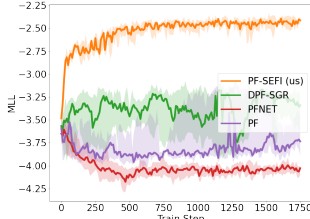

**(a)** MLL of test synthetic data with 25 step trajectories.

**(b)** MLL of test synthetic data with 50 step trajectories.

**(c)** MLL of test real data with 60 step trajectories.

**Figure 2:** Marginal Log Likelihood (MLL) on synthetic and real test data for models trained using PF-SEFI (us), DPF-SGR, PFNET, and PF, plotted against the corresponding training steps. For synthetic data we also show the MLL of the true models.

to compare the learned models against a known ground truth. We measure the quality of learned models using the following metrics:

- *Marginal Log Likelihood (MLL)*: The marginal log likelihood $\ell_T(\theta)$ given by filtering observations $y_{0:T}$ using the learned models.
- *Average Displacement Error (ADE)* and *Average Yaw Error (AYE)*: The average error in the positions and yaws respectively of the smoothed state estimates $\mathbb{E}_\theta(x_{0:T}|y_{0:T})$ against the true poses, $\bar{x}_{0:T}$. For the synthetic data, the true poses are sampled while generating the data; for the real data, the true poses are obtained by humans manually labelling object trajectories from videos. These measure the quality of the learned models for the purposes of state estimation.
- *Average Observation True Log Likelihood (AOTLL)*: The average log likelihood of observations conditioned on the true states under the learned observation model, i.e., $\frac{1}{T+1}\sum_{t=0}^{T}\log g_\theta(y_t|\bar{x}_t)$. This measures the quality of the learned observation model.
- *Average Policy True Log Likelihood (APTLL)*: The average log likelihood of true actions, $\bar{a}_{1:T}$, (only available for experiments with synthetic data since it is not possible to manually label latent actions) conditioned on the true states under the learned policy, i.e., $\frac{1}{T}\sum_{t=1}^{T}\log \pi_\theta(\bar{a}_t|\bar{x}_{t-1})$. This measures the quality of the learned policy.

**Results.** Figure 2 shows the progress of the learned models by tracking MLL of held out test data for each of the three datasets (synthetic data with 25 steps, synthetic data with 50 steps, and real data with 60 steps), and for each of the four methods (PF-SEFI, DPF-SGR, PFNET, and PF). Table 1 summarise the performance of the learned models at convergence. We pick the best hyper-parameters, smoothing lag $L$ for PF-SEFI, and trade-off parameter $\alpha$ for PFNET, in each of the experiments. Appendix C includes analysis of the training sensitivity of each of the hyper-parameters (see Figures 5, 6, and 7). We find that PF-SEFI improves with increasing $L$ up to a point, past which it is insensitive to the choice of $L$ (see Figure 6, Appendix C).

In our experiments with *synthetic data* with 25 steps (Figure 2a and Experiment A in Table 1), we observe a clear gap in performance of PF-SEFI and DPF-SGR relative to PFNET and PF. The improvements over PF are likely due to the bias in PF's score estimates due to the non-differentiable resampling step, while the improvements over PFNET are likely due to adverse effects of not resampling with the correct distribution at each time step. While PF-SEFI and DPF-SGR perform similarly on this dataset, the difference is stark in the case of synthetic data with 50 steps (Figure 2b and Experiment B in Table 1). PF-SEFI is invariant to the length of the trajectories used, converging stably; however, all other methods, struggle to learn useful models. We postulate that since each of the baselines, in one way or another, differentiate through all time steps of the filter, the variance in their score estimates is too high for good learning through gradient ascent.[1]

---

[1]The authors of DPF-SGR [17] recommend the use of stop gradients not only for particle weights after resampling, i.e., $\tilde{v}_t^i = \bar{v}_t^{a^i}/\perp\bar{v}_t^{a^i}$ [17, Algorithm 1], but also, in the case of bootstrap particle filters, while computing the likelihood ratio $v_t^i = \tilde{v}_{t-1}^i p_\theta(x_t^i, y_t|x_{t-1}^{a^i})/\perp q_\theta(x_t^i|x_{t-1}^{a^i})$ before resampling, and while sampling from $x_t^i \sim q_\theta(\cdot|x_{t-1}^{a^i})$ [17, Section 4.1]. While these additional stop gradients significantly reduce variance, our experiments with them yielded extremely poor overall performance (even with synthetic data with 25 steps). The results we report here thus make use of stop-gradients only for particle weights after resampling.

**Table 1:** Metrics computed on held out test data comparing PF-SEFI (us) against baselines. We ran experiments using 3 different datasets - (A) synthetic data with 25 steps, (B) synthetic data with 50 steps, and (C) real data with 60 steps. For experiments (A) and (B), we also compare against the performance of the true models. For experiment (C), we compare against the supervised observation model trained using manually labelled trajectories. For MLL, AOTLL, and APTLL, higher values imply better models, while for ADE and AYE, lower values imply better models.

| Exp. | Method | MLL | AOTLL | APTLL | ADE (m) | AYE (rad) |
|------|--------|-----|-------|-------|---------|-----------|
| A | TRUE | $-3.161 \pm 0.003$ | $-2.128$ | $2.674$ | $0.090 \pm 0.001$ | $0.014 \pm 0.000$ |
| | PF-SEFI (us) | $\mathbf{-3.147 \pm 0.004}$ | $-2.285 \pm 0.028$ | $\mathbf{2.661 \pm 0.014}$ | $0.186 \pm 0.021$ | $0.016 \pm 0.000$ |
| | DPF-SGR | $-3.159 \pm 0.004$ | $\mathbf{-2.265 \pm 0.010}$ | $2.594 \pm 0.027$ | $\mathbf{0.165 \pm 0.008}$ | $\mathbf{0.014 \pm 0.000}$ |
| | PFNET | $-3.225 \pm 0.004$ | $-2.487 \pm 0.026$ | $2.621 \pm 0.013$ | $0.264 \pm 0.019$ | $0.017 \pm 0.000$ |
| | PF | $-3.229 \pm 0.006$ | $-2.484 \pm 0.026$ | $2.576 \pm 0.021$ | $0.245 \pm 0.021$ | $0.017 \pm 0.000$ |
| B | TRUE | $-3.145 \pm 0.002$ | $-2.165$ | $2.693$ | $0.088 \pm 0.001$ | $0.012 \pm 0.000$ |
| | PF-SEFI (us) | $\mathbf{-3.141 \pm 0.005}$ | $\mathbf{-2.283 \pm 0.015}$ | $\mathbf{2.505 \pm 0.042}$ | $\mathbf{0.165 \pm 0.013}$ | $\mathbf{0.014 \pm 0.000}$ |
| | DPF-SGR | $-3.966 \pm 0.050$ | $-2.636 \pm 0.031$ | $0.811 \pm 0.130$ | $2.828 \pm 0.415$ | $0.142 \pm 0.016$ |
| | PFNET | $-4.169 \pm 0.046$ | $-2.901 \pm 0.039$ | $0.539 \pm 0.077$ | $2.809 \pm 0.176$ | $0.148 \pm 0.008$ |
| | PF | $-4.118 \pm 0.038$ | $-2.841 \pm 0.025$ | $0.681 \pm 0.122$ | $2.502 \pm 0.042$ | $0.137 \pm 0.007$ |
| C | SUPERVISED | N/A | $-2.224 \pm 0.006$ | N/A | N/A | N/A |
| | PF-SEFI (us) | $\mathbf{-2.447 \pm 0.029}$ | $\mathbf{-1.973 \pm 0.029}$ | N/A | $\mathbf{0.275 \pm 0.011}$ | $\mathbf{0.034 \pm 0.006}$ |
| | DPF-SGR | $-3.297 \pm 0.287$ | $-2.236 \pm 0.218$ | N/A | $0.643 \pm 0.177$ | $0.081 \pm 0.477$ |
| | PFNET | $-4.019 \pm 0.098$ | $-2.752 \pm 0.079$ | N/A | $0.746 \pm 0.091$ | $1.015 \pm 0.159$ |
| | PF | $-3.848 \pm 0.045$ | $-2.639 \pm 0.140$ | N/A | $0.701 \pm 0.109$ | $1.082 \pm 0.364$ |

The results of our experiments with *real data* with 60 steps (Figure 2c and Experiment C in Table 1) are consistent with Experiment B (i.e., with experiments on synthetic data with 50 steps) and show that PF-SEFI is able to learn useful models. The learned observation model using PF-SEFI performs even better than the model that was trained offline through supervision with manually labelled data (see AOTLL in Table 1 for Experiment C). We also find that sampling from the learned model produces observations that are qualitatively similar to the real data (Appendix D). While the supervised model is trained only on the subset of the observations that are labeled (labelling only a subset is common in practical applications due to the cost of labelling), PF-SEFI, by contrast, can leverage *all* observations in a self-supervised fashion. Moreover, we speculate that the labels contain noise and that the labelling distribution is biased towards observations that are easy to label. Both limitations hinder supervised learning.

## 6  Discussion, Limitations, and Future Work

In this work, we proposed an efficient particle-based method for estimating the score function to learn a wide class of SSMs in a self-supervised way. Compared to previous particle methods that estimate the score, PF-SEFI is more computationally efficient, allowing us to scale to learning models with many parameters. Unlike alternative methods, PF-SEFI is applicable to SSMs where the transition distribution is concentrated on a low-dimensional manifold, allowing us to apply it to a real-world AV object tracking problem. We showed empirically that our method learns better models and is more stable in training than methods that use automatic differentiation to estimate the score, and that we can learn an observation model that outperforms one trained using supervised learning.

While this solution is ideal for our problem, it does have a number of limitations. Most notably, it is restricted to maximising the marginal log-likelihood of the data, while differentiating through the filter allows for arbitrary differentiable loss functions. Furthermore, our method is not suitable for estimating the parameters of a proposal distribution. Beyond these algorithmic limitations, in our application, the models that we used were not very expressive. For the observation model, we did not model important phenomena that affect partial observability such as occlusions and we restricted our states and observations to 2D. For the policy, we used a simplified policy with only basic features that are insufficient for controlling an agent in simulation. Furthermore, the policy and the motion model are both specific to vehicles, and currently exclude other road users such as pedestrians.

In future work, we aim to scale up our problem setting, by making both models more expressive, and to estimate more state dimensions, such as full 3D poses and sizes of objects. We also believe that learning policies as components of an SSM to explicitly account for observation noise is, in practice, critical for learning good driving behaviour from demonstrations. Such policies could be used as models for predicting the behaviour of other road-users, or to control agents in simulation, and the method we proposed in this work offers an ideal starting point to explore this.

**Acknowledgments**

We thank Drago Anguelov, Charles Qi and Congcong Li for their helpful feedback on a draft of the paper. We also thank the reviewers for taking the time to give detailed and useful feedback on our initial submission. Finally, we thank the Waymo Research team and DeepMind for supporting this project.

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
