# OpenReview forum: "Particle-Based Score Estimation for State Space Model Learning in Autonomous Driving"
_robot-learning.org/CoRL/2022/Conference — CoRL 2022 Poster_

### Official Review · Reviewer_8rdy · 2022-07-15

**Originality:** Fair
**Technical Quality:** Very Good
**Clarity Of Presentation:** Good
**Impact:** 2

**Recommendation:**

Weak Reject: I recommend rejecting the paper, but will not argue for my recommendation if the majority of other reviewers have a different opinion.

**Summary:**

The model proposes training non-linear, non-Gaussian SSMs using gradients approximated with Fisher's identity. In order to efficiently compute the required smoothing distributions, it proposes using fixed-lag smoothing. The resulting method is evaluated on a state estimation task in an autonomous vehicle setting where 2D poses of surrounding vehicles need to be estimated from observations.

**Issues:**

Besides the above-mentioned points, I have minor issues:
- At first, I was a bit confused by Section 4.3, because I thought it to be a fundamental aspect of the method, not just a simplification of the dynamics model done for the specific conducted experiments (although I agree that the general method approach described in this section is applicable to a wider class of experiments). Maybe it could be considered to move this part to the experiment section.
- If I understand correctly, the MLL is computed based on the filtered estimates, while ADE and AYE use smoothed estimates. Why are different beliefs used here and are the smoothed ones "fixed-lag" using the full trajectory?

**Quality Of The Limitations Section:**

Additional details required

**Reviewer Expertise:**

4: The reviewer is confident but not absolutely certain that the evaluation is correct

**Robotics Focus:**

Relevant but unlikely to deploy to hardware in near future

**Strengths And Weaknesses:**

The paper is well written and easy to follow, and the main idea, intuitions, and mathematical details are clear.
Using fixed-lag smoothing to approximate the required smoothing estimates efficiently is a simple idea.
Yet, as this seems to be the paper's main algorithmic contribution the effects and limitations of this approximation would need further investigation. While I agree, that the assumption seems reasonable for many systems, ablations for different smoothing lengths $L$ would be good to see.
In general, I do not believe the conducted experiment is sufficient to allow assessment of the method's full potential, as it relies on unrealistic assumptions (lines 218-220), largely pre-engineered transition and observation models, and only considers a narrow scope of applications.



**Summary Of Recommendation:**

While I like the writing and simplicity of the idea, I think the experiments are not sufficient to allow assessment of the method's potential impact beyond the narrow scope it was evaluated in.

---

> ### Author Response · Authors · 2022-08-24
> **Author Response to Reviewer 8rdy**
>
> Thank you for taking the time to review our paper. We are pleased that you found our paper well-written and easy to follow. Below, we address your main comments:-
>
> > Regarding Section 4.3:-
>
> Thank you for pointing this out. Our proposed solution leveraging the fixed-lag approximation for score estimation via Fisher’s Identity (Sections 4.1-4.2) is applicable to parameter learning in a wide class of SSMs, and can therefore be used for many problems other than multi-object tracking. In this setting, the primary contribution of our algorithm over alternatives (such as [8, Algorithm 2]) is reduced computational complexity. What is interesting in multi-object tracking (and indeed, many mobile robotics applications [17]) is that the SSM’s transition distribution $f(x_t | x_{t-1})$ is concentrated in a low-dimensional manifold, because of constraints imposed by motion models. Section 4.3 demonstrates that our method is still applicable in the common setting involving a fixed motion model, while many alternatives (such as [8, Algorithm 2]) are no longer applicable. We have updated sections 1 and 6 of the paper to highlight the broader applicability of the method.
>
> > Regarding an ablation with different lag lengths:-
>
> Thank you for pointing this out. Figure 5 in the appendix of our submitted paper addressed this point but we forgot to refer to it in the main text. The figure shows the effect of different lag lengths on overall performance as measured by Marginal Log Likelihood (MLL) on the held out test sets in each of the datasets (synthetic and real). We have added a reference to these results in the main text to make this clearer. Additionally, we have further increased the number of lag lengths considered for the real dataset (Figure 5c, Appendix C and Figure 6, Appendix C) and see that if the lag is too short (of 0, 2, 3 steps) the performance is poor. However, once the lag is large enough, the effect starts to plateau. These plots helped us estimate the optimal lag, i.e., the smallest lag where the overall performance is good.
>
> > Regarding the computation of MLL using filtering estimates:-
>
> The Marginal Log Likelihood (MLL) is computed using predictive marginals at every time step. This can be derived from first principles to be the marginal likelihood of the data, i.e. $p(y_{1:T})$ [12, Section 4.1]. We additionally consider metrics such as Average Displacement Error (ADE) and Average Yaw Error (AYE) to evaluate the learned models on the real-world task of multi-object tracking in the _offline_ setting where the entire sequence of observations is available ($y_{1:T}$), and hence we used the smoothing marginals $p(x_t | y_{1:T})$ in the computation of ADE and AYE. We have now also included the use of the learned models in the _online_ setting where observations are only available leading up to a particular time step $t$ ($y_{1:t}$) by computing ADE and AYE on the filtering marginals $p(x_t | y_{1:t})$ (see Table 5 in Appendix F).
>
> > Regarding unrealistic assumptions and the narrow scope of our experiments:-
>
> We have included a number of additional experiments in Appendix E (Training on Shorter Sequences), F (Performance on the Filtering Task), G (Training with Higher Dimensional Observations), and H (Effect of a Noisy AV State on Learning). We find that PF-SEFI performs well in every one of the various circumstances considered, including when some assumptions (like perfect AV states) are not met (see Appendix H). Moreover, while we focussed our experiments on multi-object tracking in this paper, we experimented not only with synthetic data, but also with _real data_, in a complex setting with high dimensional observations. This problem is of great importance for building AVs as state estimation of objects around an AV is crucial for safe navigation. We developed this algorithm to solve an important learning problem that baselines like DPF-SGR and PFNET failed to solve. Both DPF-SGR and PFNET only report results on synthetic datasets, while we go a step further in our experiments to demonstrate the efficacy of our method on a real dataset.

---

> > ### Comment · Reviewer_8rdy · 2022-08-27
> > **Thank you for the Response**
> >
> > I would like to thank the authors for their response and acknowledge the clarifications and improvements to the paper. I will make my decision after discussion with the other reviewers.

---

### Official Review · Reviewer_ScN7 · 2022-07-31

**Originality:** Very Good
**Technical Quality:** Good
**Clarity Of Presentation:** Very Good
**Impact:** 4

**Recommendation:**

Strong Accept: I recommend accepting the paper and will argue for my recommendation even if other reviewers hold a different opinion.

**Summary:**

The paper proposes a new approach for particle filtering approach for estimating the score function of state-space models (SSM). The authors do so using the Fisher Identity to circumvent the non-differentiable sampling step in particle filtering for estimating the score function. Moreover, they circumvent the potential issue of path degeneracy where the particles converge to a single one by using a fixed lag $L$ up to which the estimates are calculated, based on the assumption that observations after a time $t+L$ are not very useful for estimates at the current time step. They also derive the use of a motion model for policy-based approximations by showing that the gradient of the policy corresponds with that of the SSM, allowing it to be plugged into the gradient of the score function. Their results for approximating the states of external objects from Bird's eye views of a vehicle show good log-likelihoods and state estimates on both real and simulated datasets.

**Issues:**

- There are some issues that I found in Lines 140-141 and Eq. 4 as $\alpha$ and $\delta$ are not defined. This needs to be cleared to properly understand the approach and how it could potentially affect the score function.


**Quality Of The Limitations Section:**

Limitations are addressed clearly

**Reviewer Expertise:**

3: The reviewer is fairly confident that the evaluation is correct

**Robotics Focus:**

Sufficient demonstration on hardware

**Strengths And Weaknesses:**

### Strengths
- The proposal for using the Fisher Identity for approximating SSMs enables low variance state estimates and also circumvents the non-differentiable nature of the sampling process.
- The computational capacity is drastically reduced by considering a fixed time window up to which the estimates are calculated rather than going through the whole trajectory.
- The paper is very well written. It is very easy to follow and the relevant works, preliminaries and approaches are explained in simple, understandable terms and the paper has a good flow from start to end.

### Weaknesses
- The paper would benefit from providing some more extensive information about the real dataset that is used, such as how the data was collected, sensors used, example data etc. It would have been better to showcase the results on some existing datasets, like KITTI for example, or even on a subset of it. This would allow the results to be more comprehensible.
- The paper is currently missing an ablation study to show the extent to which the performance is affected by the different components. These could be different Lag lengths in Sec. 4.2


**Summary Of Recommendation:**

The proposed approach is well motivated and follows a simple, intuitive and useful extension to improve the score functions of SSMs along with the added advantage of reduced computational capacity and incorporating the policy gradient in the score function estimate. The experimental evaluation shows the benefits of the proposed method on both synthetic and real data (although the real dataset is much smaller compared to traditional driving datasets like KITTI for example). Moreover, the paper is very well written and the ideas are presented well, making it a nice read.

---

> ### Author Response · Authors · 2022-08-24
> **Author Response to Reviewer ScN7**
>
> Thank you for such a detailed and insightful review! We are delighted that you found our method simple, intuitive, and useful. Below, we address your main comments:-
>
> > Regarding an ablation with different lag lengths:-
>
> Thank you for pointing this out. Figure 5 in the appendix of our submitted paper addressed this point but we forgot to refer to it in the main text. The figure shows the effect of different lag lengths on overall performance as measured by Marginal Log Likelihood (MLL) on the held out test sets in each of the datasets (synthetic and real). We have added a reference to these results in the main text to make this clearer. Additionally, we have further increased the number of lag lengths considered for the real dataset (Figure 5c, Appendix C and Figure 6, Appendix C) and see that if the lag is too short (of 0, 2, 3 steps) the performance is poor. However, once the lag is large enough, the effect starts to plateau. These plots helped us estimate the optimal lag, i.e., the smallest lag where the overall performance is good.
>
> > Regarding the lack of clarity in lines 140-141:-
>
> Thank you for pointing this out. We have updated the text and clarified the meaning of $\alpha$ and $\delta$.
>
> > Regarding details around the datasets:-
>
> The real dataset was collected by driving an AV in an urban environment, equipped with LiDARs, cameras, and radar sensors. All sensors were used to associate LiDAR points to their corresponding objects and we obtain the polygons that are visualised in Figure 1 via a convex hull computation of the associated LiDAR points. We have added these details under “Baselines, Datasets, and Metrics” in Section 5. We are also happy to add more details, for example, the areas where the AV was driven, who collected the data etc., after the review phase in order to preserve anonymity.

---

### Official Review · Reviewer_swzV · 2022-07-31

**Originality:** Good
**Technical Quality:** Good
**Clarity Of Presentation:** Very Good
**Impact:** 3

**Recommendation:**

Weak Accept: I recommend accepting the paper, but will not argue for my recommendation if the majority of other reviewers have a different opinion.

**Summary:**

This paper proposes a method for learning observation and transition models for birds-eye view multi-car autonomous driving scenarios.  Using a fixed-lag approximation of the score function along with a deterministic motion model, inference of high-dimensional models can be achieved.

**Issues:**

See above.

**Quality Of The Limitations Section:**

Limitations are addressed clearly

**Reviewer Expertise:**

3: The reviewer is fairly confident that the evaluation is correct

**Robotics Focus:**

Highly relevant to robotics but no hardware experiments

**Strengths And Weaknesses:**

Strengths:

The paper is well-written, and the method is generally well-justified.  The method leverages specific assumptions from the bird's eye view autonomous driving scenario to learn observation and transition models.

Weaknesses/Questions:

More evaluations on the fixed-lag size would be beneficial. At what point is path degeneracy an issue? At what point is a fixed-lag uninformative? Since this most likely depends on the number of particles as well, it would be nice to see a comparison with the fixed-lag window size and number of particles being modified simultaneously.

Why does the real data show less performance difference when modifying fixed-lag? Why did real data cause large gradients across all methods? If performance is independent of fixed-lag size, what is the method gaining? Plot shows minimum of L=5 for real-data, but what happens for L=1,…,5? If this small of window size gives similar results for real-data, then learning the observation and transition models may not require longer horizon inference.

If DPF-SGR performs better in terms of accuracy for 25 steps, what is preventing methods from also limiting the number of steps that is used for training?

Since other methods may or may not differentiate through the marginal log-likelihood, does it make sense to even compare the baselines using this?

Relative bearing is usually not available as a measurement, and estimates would be somewhat noisy.  Is this a realistic setup?

Other comments:

Figure 1 is somewhat wasteful in terms of white space, and also does not illustrate much.  Consider improving the figure as this is the only one in the paper.  One of the figures in the supplementary material may better illustrate the method and application than the current Figure 1.

How sensitive is the method to the agent's state noise? In reality, this will be imperfect.

Is assuming the motion model to be deterministic adequate? In real-world scenarios, this may not be the case.

**Summary Of Recommendation:**

Overall, the method provides is well-written and provides interesting methodology.  While the evaluations demonstrate the effectiveness of the method, some additional experiments demonstrating the benefits of the proposed fixed-lag approximations is needed.  One of the baselines performs better on the 25 step simulate data, while the fixed-lag size does not show much difference in performance on the real data.


Update:
The authors have addressed the main concern about the effect of the fixed-lag length, as well as other questions.  Therefore, I have changed my recommendation to weak accept.

---

> ### Author Response · Authors · 2022-08-24
> **Author Response to Reviewer swzV (Part 1 of 2)**
>
> Thank you for the thoughtful comments and insights! We are encouraged that you found our paper well-written and well-justified. Below, we address your main comments:-
>
> > Regarding the effect of small fixed-lag (L) and the tradeoff with the number of particles:-
>
> For both the synthetic and real datasets, we see a strong dependence on $L$ especially when $L$ is small. We have added $L=0,2,3,5,9,14$ in the results from our experiments with the real dataset in Figure 5c, Appendix C. These results demonstrate that $L$ needs to be of a minimal length in order to provide sufficient information to the smoothing marginals $p(x_t | y_{0:T})$. However, lags beyond 9 steps start to become uninformative and the effect plateaus. With more particles, we see more stable learning and less variance across repetitions but the plateauing effect with large values of $L$ is mostly the same (see Figure 6). While we do not observe significant effect of path degeneracy in our problem, other than some increase in variance across repetitions, we note that it is a well known problem in this space and leads to high variance gradients for large enough $T$ and $L$; see e.g. [12, Section 4.3] and [8, Figure 3]. We agree that to learn good models, empirical evidence (and theory) suggests that $L$ does not need to be too large for good parameter estimation, which also makes the computational cost of score estimation manageable.
>
> > Regarding the performance of DPF-SGR on 25 steps:-
>
> Thanks for pointing this out; it turned out to be an interesting result! We added some additional experiments training DPF-SGR and other baselines on 30 and 15 step sequences of real data (instead of 60). The results can be found in Appendix E. At 30 steps, we find that all 3 baselines still fail to learn good models (Figure 11a), while PF-SEFI performs almost as well as on length 60 sequences. At 15 steps, however, the baselines improve (Figure 11b), though their final performance is still worse than PF-SEFI on 60 steps.
>
> This highlights a big advantage of our method -  PF-SEFI is relatively invariant to the length of sequences that it’s trained on. Depending on the problem setting, there is usually a minimum sequence length required to obtain enough information to learn the correct models. If that sequence length, for a given problem setting, is longer than the maximum sequence length for which a method such as DPF-SGR is stable to train, then one must sacrifice model quality for stable learning by cutting the sequences to shorter subsequences or by subsampling the sequences, throwing away some of the observations. The additional experiments here show that trimming down to 15 steps was necessary for reasonable (though still suboptimal) models to be learned using the baseline methods. In other problems it may well be the case that even more trimming or subsampling would be needed.
>
> > Regarding the applicability of the baselines:-
>
> The baselines (DPF-SGR, PFNET, and PF) benefit from the flexibility of using targets other than Marginal Log Likelihood (MLL). However, for the purposes of parameter estimation, the standard target is the Marginal Log Likelihood as it provides consistent and efficient parameter estimates under weak regularity conditions. Moreover, baselines such as DPF-SGR use the same target for parameter estimation in their paper [17]. Hence, we believe it is fair to compare PF-SEFI to these baselines.
>
> > Regarding measuring relative bearing:-
>
> Relative bearing (in addition to range and bearing), are used as features and provided as inputs to the neural network powering the observation model. They are computed using the AV’s pose and the target object’s particles (the computation for one such particle is visualised in Figure 3a in Appendix A). They are not part of our measurements or observations.
>
> > Regarding wasteful usage of space in Figure 1:-
>
> We agree and have followed your suggestion by replacing it with more informative figures from the Appendix.
>
> > Regarding sensitivity to the error in the AV’s own state estimation:-
>
> You are right that we assume that we know the AV’s state perfectly. However, we believe that, in practice, such imperfection, especially in our setting and thanks to the suite of sensors available to the AV, is quite low. Indeed our real data is collected from a real AV, and hence already includes such noise (owing to the AV’s imperfect state estimation of itself). Since our experiments with the real data yield useful policies and observation models, we believe that PF-SEFI is robust to such misspecification. Moreover, for completeness, we ran additional experiments that provide empirical evidence of the same using synthetic data, wherein we added Gaussian noise of up to 0.5m in $x$ and $y$ and 0.05rad in $\theta$. We observed no decrease in overall performance or in learning stability. We have also included these results in Figure 12 in Appendix H of the paper.

---

> > ### Author Response · Authors · 2022-08-24
> > **Author Response to Reviewer swzV (Part 2 of 2)**
> >
> > > Regarding the assumption around the motion model:-
> >
> > While we agree that the motion model may not be perfectly deterministic, we believe that such misspecification is minor in practice and should not impede learning. In the practical case of state estimation for vehicle based road users, the application considered in this paper, unless the vehicle is under extreme circumstances such as turning at very high speeds, or skidding, we expect that the trajectories are largely explainable via the motion model considered in this paper [17]. Moreover, we note that our experiments with the real data yield solid results despite including such a potential misspecification. However, in cases where the misspecification is significant, we expect that the noise will either be explained away by the policy or by the observation model. In fact, we generated additional synthetic data containing Gaussian noise in the state on top of our motion model, and re-trained our models using PF-SEFI. The resultant policy had a higher variance relative to the true models, but training was still robust.

---

### Official Review · Reviewer_KijS · 2022-08-01

**Originality:** Good
**Technical Quality:** Very Good
**Clarity Of Presentation:** Very Good
**Impact:** 2

**Recommendation:**

Weak Accept: I recommend accepting the paper, but will not argue for my recommendation if the majority of other reviewers have a different opinion.

**Summary:**

This paper investigates particle-based state estimation under the presence of unknown observation and transition models. This is challenging for a number of reasons, in particular due to the non-differentiable resampling step in particle-based methods. Prior work has proposed leveraging Fisher’s identity to derive a maximum likelihood objective for the model parameters that bypasses the resampling step; however such existing approaches are computationally expensive for large models. This paper builds on this prior work and introduces a particle approximation that trades off bias for computations efficiency. The authors compare this approach with existing baselines and demonstrate superior performance on a real-world and synthetic AV dataset.

**Issues:**

(prefer refer to the list of weaknesses)

**Quality Of The Limitations Section:**

Limitations are addressed clearly

**Reviewer Expertise:**

2: The reviewer is willing to defend the evaluation, but it is quite likely that the reviewer did not understand central parts of the paper

**Robotics Focus:**

Highly relevant to robotics but no hardware experiments

**Strengths And Weaknesses:**

Strengths:
* This paper studies a well-motivated problem — in real world scenarios the observation / transition models may be unknown, necessitating methods that can estimate the parameters of these models.
* The introduction and methods section are well-written. The paper does a good job detailing the shortcomings of prior work, providing background information for the method, and laying out the proposed approach in a step-by-step basis.
* The experimental results are convincing, demonstrating superior performance compared to a number of baselines across a number of quantitative metrics.

Weaknesses:
* The paper highlights that this approach is useful in multi-agent settings, yet there is nothing about the method specifically tailored for multi-agent problems. While the experiments focus on a multi-agent setting, this method seems designed for any general state estimation problem. This makes the message of the paper somewhat confusing.
* Even after reviewing the appendix, the paper provides sparse details about the datasets. Ideally the authors can help to answer the following questions: who collected the real world dataset? What is the nature of the agents in the dataset? In which specific ways are the real and synthetic datasets different?
* The paper has a limited number of figures and visualizations. It would help to provide visualizations of the datasets and qualitative analysis of the results. Some of these are available in the appendix, perhaps they can be brought up to the main section of the paper.
* The limitations that the authors describe in section 6, namely scaling this method to more complex empirical domains with occlusions and high-dimensional observations. While these are not strictly necessary components, they can help to strengthen the empirical findings.

**Summary Of Recommendation:**

While this paper builds upon a body of prior work, it introduces important technical contributions that enable efficient particle-based estimation for large scale settings. The authors can improve the paper by clarifying their contribution specifically for multi-agent settings, providing more details about the datasets, providing more figures for greater interpretability, and demonstrating results on more complex domains.

**Update**: I appreciate the improvements in clarity regarding the datasets, and the new experiments on (slightly) higher dimensional settings. These are relatively minor changes, so I am keeping my initial recommendation of weak accept.

---

> ### Author Response · Authors · 2022-08-24
> **Author Response to Reviewer KijS**
>
> Thank you for the thoughtful feedback and comments! We are pleased that you found our paper well-motivated, well-written, and well-evaluated. Below, we address your main comments to strengthen the paper:-
>
> > Regarding clarity on the datasets and adding more visualisations:-
>
> We have now replaced the visualisations in Figure 1 with ones from the Appendix with more objects across many timesteps, which we believe illustrate our problem setting better. Moreover, we have clarified under “Baselines, Datasets, and Metrics” in Section 5 that the agents we consider are vehicle based road users.
>
> Synthetic data was generated using a simple heuristic policy (see Appendix B.2), the motion model described in Appendix B.1, and a generative observation model trained on real data with human labels (see Appendix A). The real dataset was collected by driving an AV in an urban environment, equipped with LiDARs, cameras, and radar sensors. All sensors were used to associate LiDAR points to their corresponding objects, and we obtain the 2D polygons shown in Figure 1 via a convex hull computation of the associated LiDAR points. We have added these details under “Baselines, Datasets, and Metrics” in Section 5. We are also happy to add more details, for example, the areas where the AV was driven, who collected the data etc., after the review phase in order to preserve anonymity.
>
> The biggest difference between the two datasets is in the respective policies. The policy used to generate the synthetic data is based on a simple heuristic as described in Appendix B.2, while the real data is generated by human drivers. Other differences include the lack of occlusions in the synthetic data, while observations in the real data include occlusions.
>
> > Regarding experiments on more complex domains and with higher dimensional observations:-
>
> While each observation (for each object around the AV at every time step) consists of 16 2D points (32 dimensions), we think our method scales well to even higher dimensional observations. In fact, we generated two more synthetic datasets with 32 and 64 2D points, and observed that training remains stable with similar results across all metrics. We have also included these findings in Table 5 in Appendix G.
>
> > Regarding clarity on our multi-agent contributions:-
>
> Indeed, our method (PF-SEFI) is quite general for parameter estimation in any State Space Model (SSM), and not limited to multi-agent settings. We mention multi-_agent_ settings in Section 1 as one example where SSMs involve complicated and difficult to specify transition models, and therefore are more amenable to learning. While we evaluate our method on a multi-object tracking problem, it is applicable to any setting that involves parameter estimation of an SSM, particularly in the challenging regime where the transition model is composed of a policy and a motion model.

---

> > ### Comment · Reviewer_KijS · 2022-08-27
> > **Thank you for your response**
> >
> > Thank you for your response. I see some meaningful improvements have been made to the paper. I will make a final decision on my recommendation after discussing with the other reviewers.

---

### Meta-Review · Area_Chair_iSz3 · 2022-09-07

**Recommendation:** Accept (Poster)
**Confidence:** 3

**Metareview:**

The authors propose a multi-object state space estimation approach based on particle filters where the gradients are computed through the Fischer's identity.

Strength:

   - Clear story and well motivated (to avoid biased or high variance gradient approximation)
   - Well-structured paper
   - Comparison to two baseline methods
   - Evaluation on a real-world dataset and two synthetic datasets
   - Detailed discussion of all model assumptions and limitations

Weakness:

   - The real-world av tracking task is not described in sufficient detail
   - Therefore it is difficult to assess how relevant the approach is and how well it could work on more complex real-world tasks
   - In general, a discussion on the applicability of the model to other problems, not only multi-object tracking, would be important

Update: The modified Figure 1 and the additional explanations at multiple places improve the paper.

**Best Paper Nomination:**

No